# Onset of Optical Instabilities in the Nonlinear Optical Transmission of Heliconical Cholesteric Liquid Crystals

Ashot H. Gevorgyan [1] and Francesco Simoni [2,3,*]

1    School of Natural Sciences, Far Eastern Federal University, 10 Ajax Bay, Russky Island, 690922 Vladivostok, Russia; agevorgyan@ysu.am
2    Dipartimento SIMAU, Università Politecnica delle Marche, 60131 Ancona, Italy
3    Institute of Applied Sciences and Intelligent Systems of CNR, 80072 Pozzuoli, Italy
*    Correspondence: f.simoni@photomat.it

**Abstract:** We report the result of an accurate calculation of optical transmission of a light beam traveling along the helix direction of a heliconical cholesteric liquid crystal when the light intensity is affecting the director orientation by an effective optical torque. The study is based on the application of Ambartsumian's layer addition modified method with the aim of taking into account the continuous modification of the wave field during propagation, which is quite strong when the light wavelength approaches the Bragg resonance. We show first, that the recently calculated red shift of the transmission gap is confirmed under the constant intensity approximation. Additionally, by taking into account the full modulation of the optical field occurring at wavelengths close to the Bragg resonance, a weak red shift is already observable at very low intensities, but quickly has the onset of instabilities in the optical transmission. We give the first account of this effect, which is dependent on the light intensity showing and that it corresponds to the onset of non-uniform distribution of the conical angle and pitch of the structure.

**Keywords:** heliconical liquid crystals; nonlinear optics; pitch tuning; optical instabilities

## 1. Introduction

Light propagation through the helical structure of cholesteric liquid crystals (CLC) has been studied for a long time, and a large amount of literature produced on this subject has been well referenced in different books on liquid crystals [1–4]. Besides the explanation of the strong Bragg reflection observed in these materials, the theoretical approaches aimed at accounting for the optical behavior under different geometrical or physical conditions can be a change of light incidence angle, change of temperature, application of electric field, etc.

After the discovery of the giant optical nonlinearity (GON) in nematic liquid crystals (NLC), the change of the optical properties of a CLC induced by light traveling through the material has been considered. However, these effects demonstrated to be very weak with respect to the ones occurring in NLC [5,6] due to the twist deformation that strongly affects the polarization of the light beam. Specifically, no significant tuning of the Bragg resonance has been observed due to light-induced reorientation in pure CLC.

From this point of view, heliconical cholesteric liquid crystals (ChOH-oblique helicoidal cholesterics) provide a new phenomenology linked to the peculiar conical arrangement of the molecular director. This configuration occurs in CLC when the bend elastic constant $K_3$ is lower than the twist elastic constant $K_2$, that is $K_3 < K_2$, and an applied low-frequency electric field (denominated "static" in the following) is applied within a range of values between two critical fields $E_{N^*C} < E < E_{NC}$, in a direction parallel to the helix axis. In this case, when $E < E_{N^*C}$ we have a conventional CLC structure with a helix axis rotated by 90° with respect to the conical configuration, while for $E > E_{NC}$, we have a complete unwinding of the structure that becomes a uniaxial nematic [7–9]. ChOH were predicted more than fifty years ago by De Gennes [10] and Meier [11] and were realized a

few years ago by the team led by O. Lavrentovich [7] using novel dimeric liquid crystals with two rigid rod-like units connected by a flexible chain with an odd number of links. Then in ChOH, the molecular director $\mathbf{n} = (\sin\theta\cos\phi,\ \sin\theta\sin\phi,\ \cos\theta)$ rotates in space on a conical surface, forming an angle $0 < \theta < 30°$ with the helix axis (here chosen as the **z** axis); therefore, the azimuthal angle is periodically modulated, $\phi(z) = \left(\frac{2\pi}{P}\right)z = qz$, where $p$ is the helix pitch.

Different from CLC, the bend deformation present in ChOH allows a strong coupling between the optical field of a light beam traveling through this structure, and the molecular director **n** gives rise to an optical torque, such as the one that originates GON in nematic liquid crystals. This effect has been experimentally demonstrated by the observation of light-induced tuning of the Bragg resonance vs. the impinging intensity using a fixed value of the static electric field [12]. Since the optical torque acts in the opposite direction with respect to the static field torque, its action corresponds to an increase in the conical angle that is to an increase of the helix pitch. The result is a weakening of the effect of the static field necessary to stabilize the conical structure; in other words, the light reduces the effective field $E_{eff}$ that determines the conical angle $\theta$ and the pitch $p$ of the ChOH structure.

In a recent paper [13], the nonlinear effects due to this optical reorientation on the light propagation of a light beam have been highlighted. In fact, if the light wavelength is close to the Bragg reflection band, the light-induced shift of the resonance itself changes the traveling conditions of the light beam increasing or reducing reflectivity dependent on intensity and polarization. This leads to a strongly nonlinear behavior in light transmission with the appearance of stop bands for a range of values of the light intensity that is dependent on the value of the applied static field. In that paper, the strong approximation of keeping the intensity constant has been remarked as the main limit of the obtained results and the need for a more accurate calculation has been underlined.

Here, we consider the same problem discussed in ref. [13] by performing a more accurate calculation that allows accounting for the nonlinear feedback of the structure parameters on the light propagation. The method has been used previously for different studies related to light propagation in CLC and is based on Ambartsumian's layer addition modified method [14]. This method makes it possible not only to calculate the components of the reflected and transmitted fields but also the field inside the system and, therefore, to study the features of the localization of light in the system. In this way, the sample is considered as a stratified material and the optical field is recalculated after traveling through each layer, allowing a more careful determination of the effects of the light field on the structure and, as a consequence, of the light transmitted by the sample.

We show first that under the constant intensity approximation, this method confirms the red shift of the transmission gap recently calculated [13]. Then, by taking into account the full modulation of the optical field occurring at wavelengths close to the Bragg resonance, a weak red shift is already observable at very low intensities; however, very soon we have the onset of instabilities in the optical transmission. We give here, the first account of this effect that is dependent on the light intensity and show that it corresponds to the onset of non-uniform distribution of the conical angle and pitch of the structure.

## 2. Theoretical Approach and Calculation Method

The effect of the optical field on the liquid crystal structure is considered by using the Meier's theory [12]. The free energy density in Gauss units is written as:

$$f_K + f_E + f_{OPT} = \frac{1}{2}K_2\left[q_0 + \sin^2\theta\left(\frac{\partial\phi}{\partial z}\right)\right]^2 + \frac{1}{2}K_3\sin^2\theta\cos^2\theta\left(\frac{\partial\phi}{\partial z}\right)^2 - \frac{\Delta\epsilon}{8\pi}(\mathbf{n}\cdot\mathbf{E}_S)^2 - \frac{\Delta\epsilon_{OPT}}{16\pi}(\mathbf{n}\cdot\mathbf{E}_{OPT})^2 \qquad (1)$$

This equation is obtained assuming:

$$\theta = const. \qquad then \quad \frac{\partial\theta}{\partial z} = 0 \qquad ; \qquad and \qquad \frac{\partial}{\partial x} = \frac{\partial}{\partial y} = 0$$

The "static" field $\mathbf{E}_S$ is oriented along the **z** axis chosen in the direction of the helix axis:

$$\mathbf{E}_S = E_S \hat{k}$$

We also assume negligible the effect on the azimuthal angle $\phi$ based on the previous work on planar cholesterics [5,6].

Equation (1) becomes:

$$f = \frac{1}{2} K_2 \left[ q_0 + \sin^2\theta \left( \frac{\partial \phi}{\partial z} \right) \right]^2 + \frac{1}{2} K_3 \sin^2\theta \cos^2\theta \left( \frac{\partial \phi}{\partial z} \right)^2 - \frac{\Delta \epsilon}{8\pi} E_S^2 \cos^2\theta - \frac{\Delta \epsilon_{OPT}}{16\pi} \sin^2\theta \, A_{OPT}^2 \qquad (2)$$

where $A_{OPT}^2$ is the effective square of the optical field acting on the liquid crystal director (see below). Here, $\Delta \epsilon$ is the low frequencies dielectric anisotropy and $\Delta \epsilon_{OPT}$ is the dielectric anisotropy at optical frequencies.

By neglecting the term $-(\Delta \epsilon_{OPT}/16\pi)A_{OPT}^2$ not dependent on $\theta$, the free energy density is rewritten as:

$$f = \frac{1}{2} K_2 \left[ q_0 + \sin^2\theta \left( \frac{\partial \phi}{\partial z} \right) \right]^2 + \frac{1}{2} K_3 \sin^2\theta \cos^2\theta \left( \frac{\partial \phi}{\partial z} \right)^2 - \left( \frac{\Delta \epsilon}{8\pi} E_S^2 - \frac{\Delta \epsilon_{OPT}}{16\pi} A_{OPT}^2 \right) \cos^2\theta \qquad (3)$$

Minimization of Equation (3) is performed to find the value of the equilibrium conical angle $\theta$ in the same way as is done in absence of the optical field. By comparing the two cases, it is straightforward to see that now the equilibrium is determined by an effective field $E_{eff}$ given by:

$$E_{eff} = \sqrt{E_S^2 - \frac{\Delta \epsilon_{OPT}}{2\Delta \epsilon} A_{OPT}^2} \qquad (4)$$

Using the light intensity $I$ we can write the effective field as:

$$E_{eff} = \sqrt{E_S^2 - \frac{\Delta \epsilon_{OPT}}{\Delta \epsilon} \frac{4\pi}{n_{av} c} I} \qquad (5)$$

where $n_{av}$ is the average refractive index, $c$ is the speed of light in vacuum.

Therefore, according to the previously reported results on ChOH [7–9], we are able to evaluate $\theta(I)$ and the pitch $p(I)$ from the known dependences of $\theta$ and $p$ on the applied field:

$$\sin^2\theta = \frac{\kappa}{1 - \kappa} \left( \frac{E_{NC}}{E_{eff}} - 1 \right) \quad \text{and} \quad p = \kappa \frac{E_{NC} p_0}{E_{eff}} \qquad (6)$$

where $\kappa = K_3/K_2$ and $p_0$ is the pitch of the correspondent planar cholesteric.

Under these conditions light propagation is investigated in the following way: we consider light propagation through the ChOH layer as having the dielectric permittivity and magnetic permeability tensors in the forms:

$$\hat{\varepsilon}(z) = \begin{pmatrix} \varepsilon_m + \frac{\Delta \varepsilon_{eff}}{2} \cos(2qz) & \frac{\Delta \varepsilon_{eff}}{2} \sin(2qz) \\ \frac{\Delta \varepsilon_{eff}}{2} \sin(2qz) & \varepsilon_m - \frac{\Delta \varepsilon_{eff}}{2} \cos(2qz) \end{pmatrix}, \; \hat{\mu}(z) = \hat{I} \qquad (7)$$

where $\Delta \varepsilon_{eff} = \varepsilon_{eff} - \varepsilon_\perp$, $\varepsilon_{eff} = \frac{\varepsilon_\perp \varepsilon_\parallel}{\varepsilon_\parallel - (\varepsilon_\parallel - \varepsilon_\perp)\sin^2\theta}$, $\Delta \varepsilon_{opt} = \varepsilon_\parallel - \varepsilon_\perp$, where $\varepsilon_\parallel$ and $\varepsilon_\perp$ are the dielectric permittivity parallel and perpendicular to the director in the nematic phase. As usual, $q = 2\pi/p$. Then, we divide the ChOH layer of thickness d into a large number of thin sublayers of thicknesses $d_1, d_2, d_3, \ldots, d_L$. If the maximal thickness is small enough, we

can assume that the optical characteristic parameters are constant in each sublayer. Then the problem is reduced to the solution of the following system of matrix difference equations:

$$\hat{R}_m = \hat{r}_m + \widetilde{\hat{t}}_m \hat{R}_{m-1} \left( \hat{I} - \widetilde{\hat{r}}_m \hat{R}_{m-1} \right)^{-1} \hat{t}_m,$$
$$\hat{T}_m = \hat{T}_{m-1} \left( \hat{I} - \widetilde{\hat{r}}_m \hat{R}_{m-1} \right)^{-1} \hat{t}_m, \tag{8}$$

with initial conditions $\hat{R}_0 = \hat{0}$ and $\hat{T}_0 = \hat{I}$ [14]. Here, $\hat{R}_m$, $\hat{T}_m$, $\hat{R}_{m-1}$ and $\hat{T}_{m-1}$ are the reflectance and transmittance matrices of the system with $m$ and $(m-1)$ sublayers, respectively; $\hat{r}_m$, $\hat{t}_m$ are the reflectance and transmittance matrices of the $m$-th sublayer; $\hat{0}$ is the zero matrix; $\hat{I}$ is the unit matrix, and the respective matrices for the reverse light propagation are denoted by tilde. The field in the medium will be calculated by the method presented in [14]. We will calculate the reflection coefficient $R = \frac{|E_r|^2}{|E_i|^2}$, transmission coefficient $T = \frac{|E_t|^2}{|E_i|^2}$, and light intensity of the total wave excited in the ChOH layer $I(z) = |E_{in}(z)|^2 I_i$ for the diffracting eigen polarization (EP). EP is the polarization of the incident light, which does not change upon transmission through the system. This polarization coincides with the polarization of one of the eigenmode, practically with the circular polarization with the same handedness of the ChOH helix. Here, $E_i$, $E_r$ and $E_t$ are the fields of the incident, reflected, and transmitted waves, respectively and $I_i$ is the intensity of incident light. In the linear limit, we will take $I_i = I_0 = 1$, and in general $I_i = N I_0$. As is well known, (see, in particular [15] and references therein) the function $I(z)$ in a CLC layer has complex behavior and it can be either much higher or much lower than the intensity of the incident light. Then, we investigate how the intensity $I(z)$ affects the structure of the ChOH layer and how can this effect be taken into account in the light propagation. To this aim, we organize our calculations as follows. In the first step ($j = 1$) we take:

$$E_{eff} = \sqrt{ E_S^2 - \frac{\Delta\epsilon_{OPT}}{\Delta\epsilon} N } = \text{const}, \quad p = \text{const and } s = \sin^2\theta = \text{const} \tag{9}$$

With these parameters we calculate the reflection and transmission coefficients as well as the light localization in the ChOH layer, that is the $I(z, \lambda) = |E_{in}(z, \lambda)|^2 I_i$. In the second step ($j = 2$) we take $E_{eff}(z, \lambda) = \sqrt{ E_S^2 - \frac{\Delta\epsilon_{OPT}}{\Delta\epsilon} |E_{in}(z, \lambda)|^2 N }$ and take into account its influence on helix pitch and angle $\theta$. Therefore, we will have $p(z, \lambda, j) = \kappa \frac{E_{NC} p_0}{E_{eff(z,\lambda,j)}}$ and $s(z, \lambda, j) = \sin^2\theta = \frac{\kappa}{1-\kappa} \left( \frac{E_{NC}}{E_{eff(z,\lambda,j)}} - 1 \right)$. Now, we calculate the reflection and transmission coefficients as well as light localization in the ChOH layer, that is the $I(z, \lambda, j) = |E_{in}(z, \lambda, j)|^2 N$ for these new parameters, and so on with $j = 3, 4, 5, \dots$

The parameters chosen for the calculation are the same as ref. [13]: $\varepsilon_{\parallel} = 2.79$, $\varepsilon_{\perp} = 2.19$, $\Delta\varepsilon = 4.79$, $\kappa = 0.1$, the unperturbed helix pitch is $p_0 = 1400$ nm, the layer thickness is $d = 20$ μm with critical fields $E_{NC} = 4.88$ V/μm and $E_{N*C} = 1.53$ V/μm and we take $n_{av} = \sqrt{(\varepsilon_{\parallel} + \varepsilon_{\perp})/2}$, where $n_{av}$ is the refractive index of the ChOH layer. The static field applied along helix direction is $E_s = 2.03$ V/μm. In our calculations, all the above mentioned sublayers are the same: $d_1 = d_2 = d_3 = \dots = d_L = 4$ nm.

In conclusion of this section, we remark that all programs were compiled in Visual Basic (based on analytical expressions for the reflection and transmission matrices for an anisotropic layer [16] and the layer addition method [14]) and debugged by the authors, and the graphs were built using the Excel and MATLAB programs.

## 3. Results and Discussion

Firstly, we check our approach by considering negligible the change in the intensity through the sample and using Equation (9) with fixed values of intensities in all the sublayers. The obtained transmission spectra are shown in Figure 1, where the transmission

$T$ vs. the light wavelength $\lambda$ is reported for increasing values of the intensities. These data coincide with what was reported in [13].

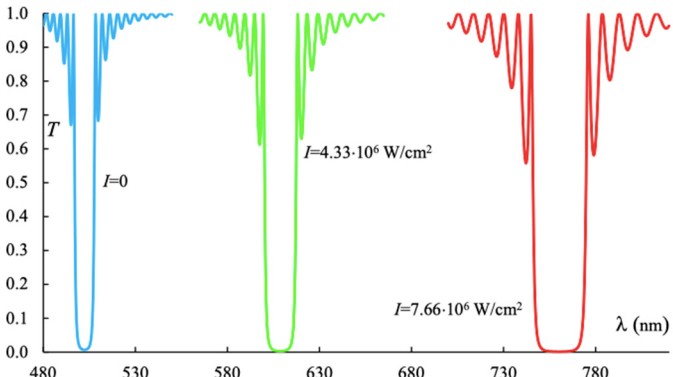

**Figure 1.** Transmission spectra of a ChOH sample for the considered diffracting eigenmode for increasing values of the light beam intensity. The static field is $E = 2.03$ V/μm. Material parameters are the ones reported in the text. The chosen intensities are the same as in ref. [13]: $I_0 = 0.0$ W/cm$^2$, $I_1 = 4.33 \cdot 10^6$ W/cm$^2$, $I_2 = 7.66 \cdot 10^6$ W/cm$^2$.

By dropping this strong approximation, we allow a recalculation of the impinging light field according to the approach described in the former section. Starting with a very low value of the incident intensity ($N = 1$ that is $I = 3.8 \cdot 10^2$ W/cm$^2$). In this case, the result is very stable and increasing the iteration parameter $j$ does not significantly change the final transmission of the light, as shown in Figure 2, for wavelengths around the Bragg resonance. Here, it is reported the transmission spectrum (a) at $j = 1$, and (b) the spectrum of the change in transmission $\Delta T = T(j = 2) - T(j = 1)$ for $N = 1$. In this case, the influence of the total field intensity excited in the medium is very small, being below 1% at the maximum variation occurring at the edge of the Bragg stop band, as observed in Figure 2b.

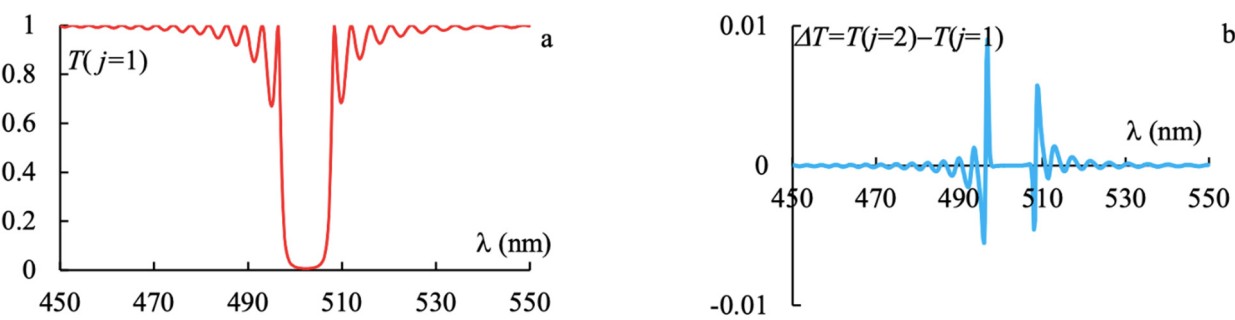

**Figure 2.** The transmission spectrum (**a**) at $j = 1$ and (**b**) spectrum of $\Delta T = T(j = 2) - T(j = 1)$ in the case $N = 1$.

Figure 3 shows the transmission spectra for four increasing values of the light intensity correspondent to increasing values of the parameter $N$ ($N = 400$ i.e., $I = 1.52 \cdot 10^5$ W/cm$^2$; $N = 750$ i.e., $I = 2.85 \cdot 10^5$ W/cm$^2$; $N = 1000$ i.e., $I = 3.8 \cdot 10^5$ W/cm$^2$; $N = 5000$ i.e., $I = 1.9 \cdot 10^6$ W/cm$^2$).

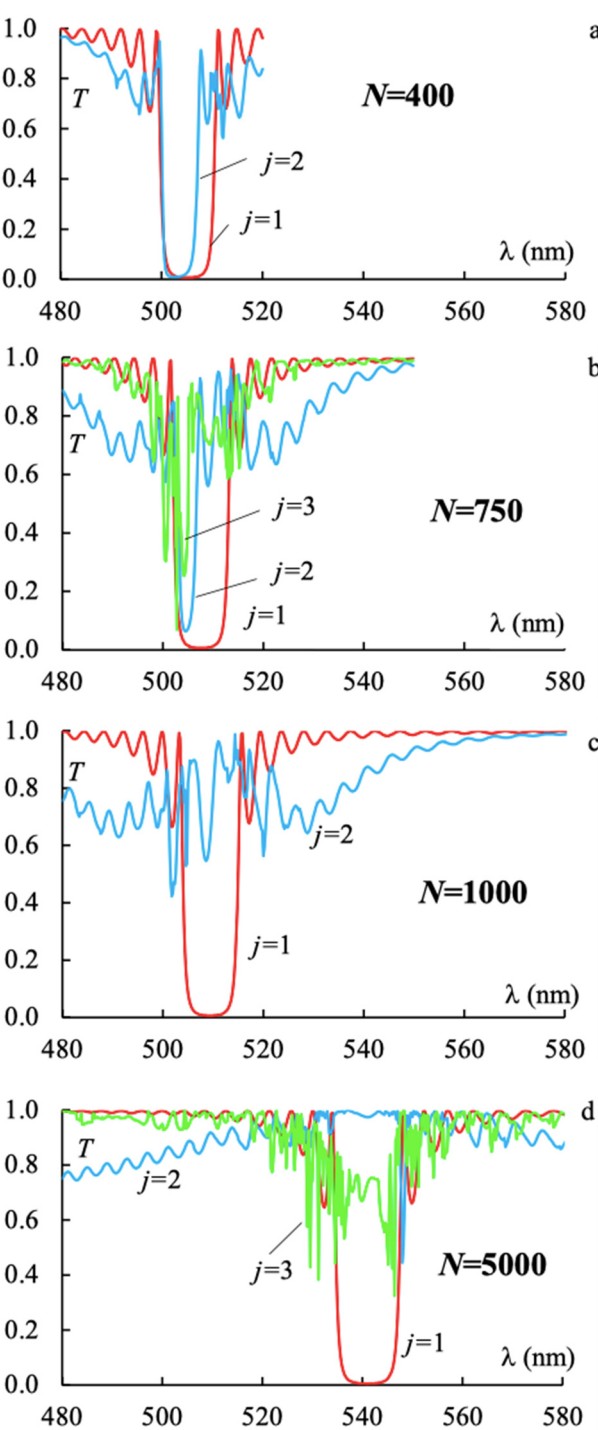

**Figure 3.** The transmission spectra for different intensities of incident light (for different $N$) and at different $j$. (**a**) $I = 1.52 \cdot 10^5$ W/cm$^2$; (**b**) $I = 2.85 \cdot 10^5$ W/cm$^2$; (**c**) $I = 3.8 \cdot 10^5$ W/cm$^2$; (**d**) $I = 1.9 \cdot 10^6$ W/cm$^2$.

As expected, at these intensities we observe (about one order of magnitude lower than the ones considered in the plot of Figure 1) a very weak red shift in the Bragg resonance (about 5 nm from the lower and the higher of the considered intensities) at the first step of calculation ($j = 1$). However, differences arise when increasing the $j$ value. In the case of $N = 400$, we observe a slight adjustment of the Bragg resonance location and width, which will become stable after some additional steps. In the other cases, the situation is very different: by increasing the $j$ parameter, the clear Bragg resonance transforms into an irregular spectrum, and it happens more quickly as the intensity increases.

It was not possible to perform the calculation of the full spectrum for higher and higher values of the *j* parameters because of the extremely high computer time requested. However, in order to give a first account of this effect, we have performed the calculation at a single wavelength close to the edge of the Bragg stop band. Thus, we are able to demonstrate the actual occurrence of the instability. This is shown in Figure 4, where the transmission *T* vs. *j* is reported for different values of the intensity of the incident light and the fixed value of $\lambda = 490$ nm.

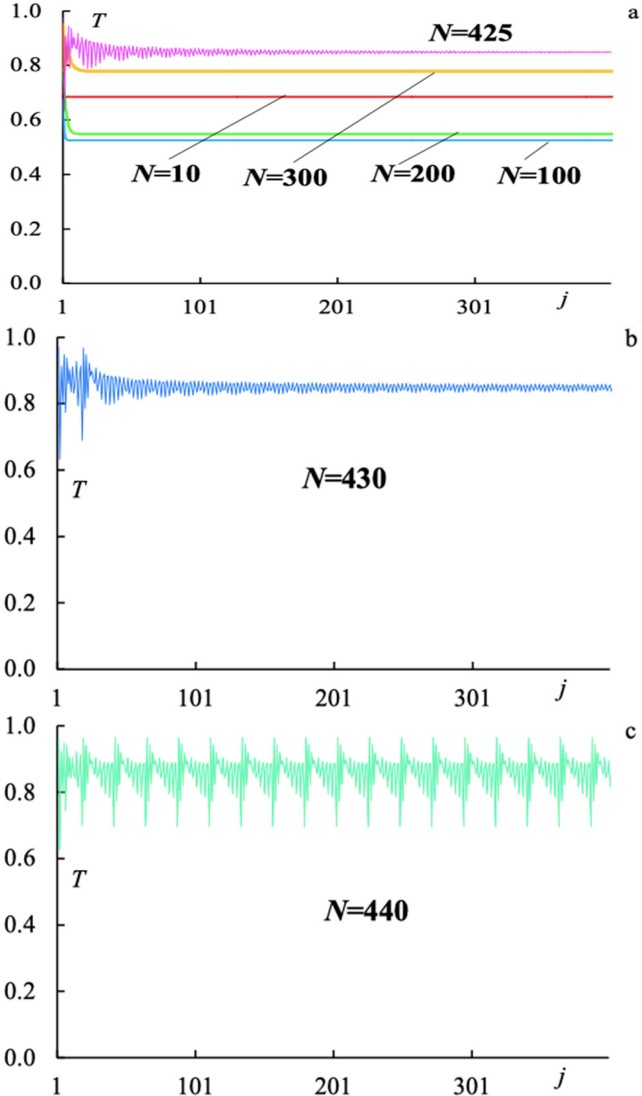

**Figure 4.** The dependence of the transmission *T* vs. *j* for different values of intensity of the incident light. Light wavelength $\lambda = 490$ nm. (**a,b**) correspond to intensities leading to a steady state value of transmission; (**c**) transmission at intensity just above the critical value to get undamped oscillations.

Looking at Figure 4a, we observe that with a very low intensity ($N = 10$ i.e., $I = 3.8 \cdot 10^3$ W/cm$^2$) no change occurs, while increasing the intensity by one order of magnitude after a few steps, a steady-state value is reached, shown by the flat lines. For *N* values just above 400 ($I = 1.52 \cdot 10^5$ W/cm$^2$), we observe a slower and slower approach to the steady-state value achieved for $j > 300$ at $N = 425$ ($I = 1.62 \cdot 10^5$ W/cm$^2$) through irregular oscillations whose amplitude decreases with increasing *j*. From Figure 4b,c, we observe a transition from damped to undamped oscillations in the light transmission. With the used parameters, this occurs at a critical value of $N_{cr}$, such that $430 < N_{cr} < 440$.

In order to correlate the observed instabilities in the optical transmission to the orientational conformation of the ChOH layer, we have calculated the function given in Equation (6). Figure 5 shows the density plots of the $s = sin^2\theta$ and helix pitch $p$ versus $z/p_0$ and $j$. For two values of the intensities, one below ($N = 400$ i.e., $I = 1.52 \cdot 10^5$ W/cm$^2$) and one above ($N = 1000$ i.e., $I = 3.8 \cdot 10^5$ W/cm$^2$) the critical value leading to un-dumped oscillations. At $N = 400$, after a sharp change in the angle $\theta$ and in the pitch of the helix at small values of $j$, a regular and small modulation of these quantities is established over the ChOH layer, and it does not change by increasing $j$. At $N = 1000$, irregular changes in the angle $\theta$ and in the pitch of the helix are observed, both through the layer thickness $z$ and vs. $j$. Therefore, by increasing the intensity, the occurrence of a strong instability in the heliconical structure is established.

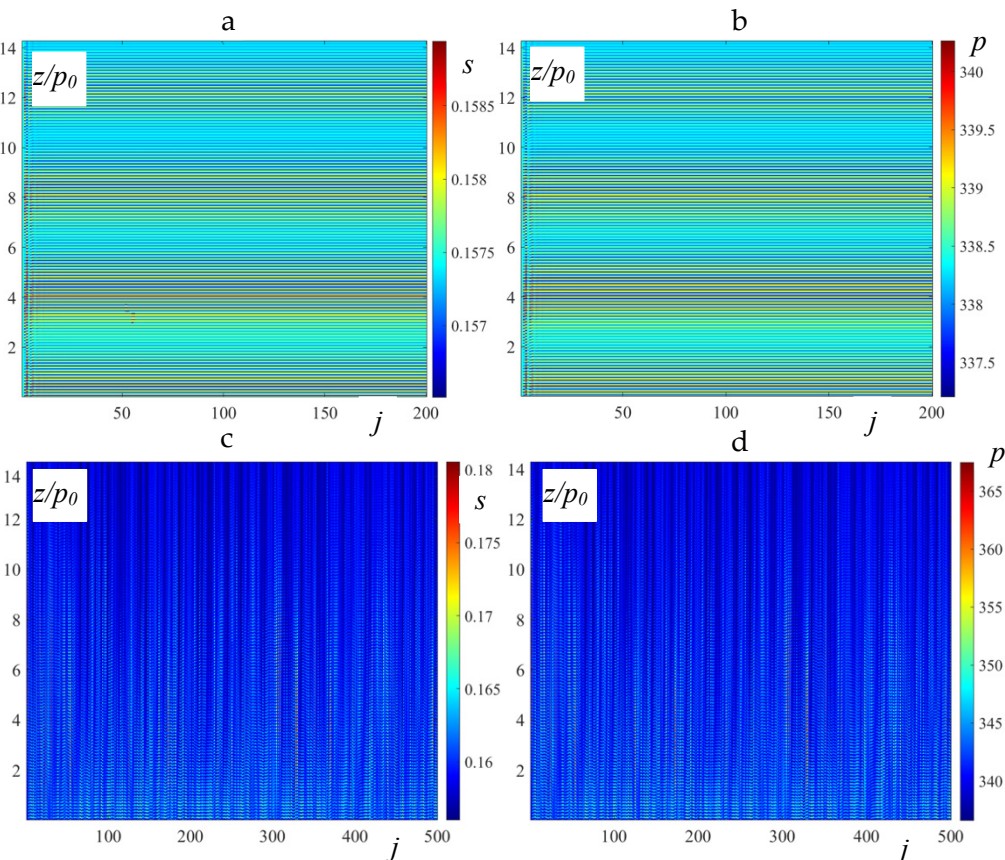

**Figure 5.** The density plots of $s = sin^2\theta$ (**a**) and helix pitch $p$ (**b**) versus $z/p_0$ and $j$ at $N = 400$. The same quantities plotted at $N = 1000$ (**c**,**d**).

## 4. Conclusions

Using Ambartsumian's layer addition modified method, we have studied the nonlinear light propagation of a light beam along the helix direction of a heliconical cholesteric liquid crystal, when the nonlinearity arises from light-induced optical torque on the molecular director. Thus, it is possible to take into account the continuous modification of the wave field during propagation, which is large when the light wavelength approaches the Bragg resonance. We first show that the recently calculated red shift of the transmission gap is confirmed under the constant intensity approximation [13]. However, by considering the full modulation of the optical field occurring at wavelengths close to the Bragg resonance, after a weak red shift already present at very low intensities, we have shown the onset of instabilities in the optical transmission. We have pointed out that a critical intensity exists dependent on the material parameters for the onset of an oscillating behavior in the optical transmission for wavelength close to the edge of the Bragg resonance. Moreover, we have demonstrated that such light intensity instabilities correspond to non-uniform distribution

of the conical angle and pitch of the structure. Concerning the previous analytical discussion of this problem [13], our results confirm that those findings can be correctly applied to a pump/probe configuration where the exciting beam has a wavelength far from the Bragg resonance that keeps its intensity nearly constant through the sample, and the probe beam with low intensity is at the edge of the Bragg resonance. On the contrary, a more complex phenomenology applies to a single beam traveling in the medium with an intensity high enough to induce significant optical reorientation and wavelength near the initial location of the Bragg wavelength of the helical structure.

**Author Contributions:** Conceptualization, A.H.G. and F.S.; theoretical approach, F.S.; numerical calculations, A.H.G.; data analysis, A.H.G. and F.S.; writing—review and editing, A.H.G. and F.S. All authors have read and agreed to the published version of the manuscript.

**Funding:** This research received no external funding.

**Institutional Review Board Statement:** Not applicable.

**Informed Consent Statement:** Not applicable.

**Data Availability Statement:** Data available at A.H.G.

**Acknowledgments:** A.H.G. acknowledges the Foundation for the Advancement of Theoretical Physics and Mathematics "BASIS" (Grant No. 21-1-1-6-1).

**Conflicts of Interest:** The authors declare no conflict of interest.

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
