# Peer review of "Onset of Optical Instabilities in the Nonlinear Optical Transmission of Heliconical Cholesteric Liquid Crystals"

_photonics, doi:10.3390/photonics9030139_

Round 1

Reviewer 1 Report

The manuscript titled “Onset of optical instabilities in the nonlinear optical transmission of heliconical cholesteric liquid crystals”, authored by A. H. Gevorgyan and F. Simoni, describes the theoretical calculations regarding the optical properties of a special type of cholesteric liquid crystals, in which the director of the orientational order is not perpendicular to the helix axis. The numerical calculations indicate the presence of instabilities above a certain critical value of incident light intensity. The results look reliable and they are within the scope of the Photonics journal. However, there are a few details which I find confusing and before acceptance, I would like to see the Authors’ comments.

line 37 - I would ask the Authors to explain explicitly the ChOH notation, because it does not look like as a simple abbreviation of heliconical cholesteric liquid crystals (it means Cholesterics Oblique Helicoidal, I suppose).

line 50 - “an angle ? > 0 with the helix axis” - shouldn’t it be written 30° > θ > 0, as it was given in Ref. 12? I understand that the Authors use the same definition of the θ angle as it is used in Figure 1 in Ref. 12.

lines 58-59 - “…its action corresponds to an increase of the conical angle that is to an increase of the helix pitch” - I find it interesting that increase of the conical angle leads to increase of the helix pitch. At the first sight, one can say that increasing of θ leads to flattening of the cone and consequently, to the decrease of the helix pitch. Of course, according to Equations (6), the pitch increases with increasing θ, but I find this result quite unintuitive. Could the Authors make a comment on this?

Equations (2,3) - I understand that in the last term, the Authors use the identity -A2sin2θ = -A2 + A2cos2θ, however, what happened to the -A2 term?

line 129 - There is a typo in “propagation”.

line 134 - There is a typo in “nematic”

Some information of the program used for numerical calculations should be included in the paper, to enable easier reproduction of similar results by other researchers (according to the Aims & Scope of the journal).

Reviewer 2 Report

The authors theoretically investigated the interaction between helicoidal cholesteric liquid crystals and light. They found instability that had not been found before.

This work is an extension of a previous study by one of the authors. However, the conclusion is very different and novel. The predictions made by theoretical studies motivate us to conduct experiments, and we believe that this paper is worthy of publication.

Basically, there is no need for revision, but I would like to point out the following points that caught my attention.

1, Figure 1

The authors used red, blue, and green lines for the short- middle- and long-wavelength spectra regions. I recommend the authors use colors corresponding to wavelength.

2,Page6, Line 1

for three increasing -> for four increasing?

3, Figure 3

Please align the length of the vertical axis in the figures a to d.

Reviewer 3 Report

I enjoyed reading this manuscript. I envision it will be of a high interest to researchers working in the field of the liquid crystals applied optics.

I recommend publishing the manuscript in its present form.

Author Response

Thank you for the reviewer's comment